# Challenges of soil carbon sequestration in NENA Region

**Darwish[1,3*], Talal; Atallah[2], Therese; and Ali Fadel[1]**

  **1.** National Council for Scientific Research, Beirut, Lebanon
  **2.** Faculty of Agricultural and Veterinary Sciences, Lebanese University
  **3.** Intergovernmental Technical Panel on Soil (ITPS)
     * Corresponding author: tdarwich@cnrs.edu.lb

Abstract

Nearorth East North Africa (NENA) region spans over 14% of the total surface of the Earth and hosts 10% of its population. Soils of the NENA region are mostly highly vulnerable to degradation, and food security will depend much on sustainable agricultural measures. Weather variability, drought and depleting vegetation are dominant causes of the decline in soil organic carbon (SOC). In this work the situation status of SOC was studied, using a land capability model and soil mapping. The land capability model showed that most NENA countries (17 out of 20), suffer from low productive lands (>80%). Stocks of SOC were mapped (1:5 Million) in topsoils (0-30 cm) and subsoils (30-100 cm). The maps showed that 69% of soil resources present a stock of SOC below the threshold of 30 tons ha$^{-1}$. The stocks varied between $\approx$ 10 tons ha$^{-1}$ in shrublands and 60 tons ha$^{-1}$ for evergreen forests. Highest stocks were found in forests, irrigated crops, mixed orchards and saline flooded vegetation. The stocks of soil inorganic carbon (SIC) were higher than those of SOC. In subsoils, the SIC ranged between 25 and 450 tons ha$^{-1}$, against 20 to 45 tons ha$^{-1}$ for SOC. This paper also highlights the modest contribution of NENA region to global SOC stock in the topsoil not exceeding 4.1%. The paper also discusses agricultural practices that are favorable to carbon sequestration. Practices of conservation agricultureOrganic amendment, no till or minimum tillage, crop rotation, mulching could be effective, as the presence of soil cover reduces the evaporation, water and wind erosions. Further, the introduction of legumes, as part of a cereal-legume rotation, and the application of nitrogen fertilizers to the cereal, caused a notable increase of SOC after 10 years. The effects of crop rotations on SOC are related to the amounts of above and belowground biomass produced and retained in the system. Some knowledge gaps exists especially in aspects related to the impact of climate change and effect of irrigation on SOC, and on SIC at the level of soil profile and soil landscape. Still, major constraints facing soil carbon sequestration are policy relevant and socio-economic in nature, rather than scientific.

Keywords: Drylands, soil organic carbon, soil inorganic carbon, land capability, C stock, conservation practices.

## 1. Introduction

The Near East North Africa (NENA) region spans over 14% of the total surface of

the Earth and hosts 10% of its population (Elhadi, 2005). The largest importer of wheat in

the world, this region is also one of the poorer (FAO, 2015). A recent assessment of global

hunger index (GHI), based on four indicators -undernourishment, child wasting, child

stunting, and child mortality- showed that most of the NENA countries present low to

moderate GHI. Countries suffering from armed conflicts, Syria, Iraq and Yemen, are at a

serious risk (von Grebmer et al., 2017). With the scarce natural resources and difficult socio-economic conditions, it is questionable whether food security will be reached by 2030, unless a significant change in agricultural practices and governance occurs (FAO, 2017).

Most of the land area of the NENA region falls in the hyper-arid, arid and semi-arid climatic zones. Climate change is expected to exacerbate the scarcity of water and drought effect. Weather variability, drought and depleting vegetation are major concerns in the loss of soil productivity and agricultural sustainability. Instabilities in SOC can affect the density of greenhouse gases in the atmosphere and negatively affect the global climate change (Lal, 2003). In fact, destructive land management practices are impacting soil functions. Land use change, mono-cropping and frequent tillage are considered to cause a rapid loss of SOC (Guo et al., 2016). These agricultural practices disrupt the stability of inherited soil characteristics, built under local land cover and climate (Bhogal et al. 2008). Thus, most NENA lands contain ~1 % of SOC, and frequently less than 0.5%.

Despite the constraints of NENA pedo-climatic conditions, increasing SOC levels is critical and challenging (Atallah et al., 2015). To maintain soil productivity and land quality, several technical and socio-economic measures need to be adopted. Additional efforts oriented to maintaining and increasing SOC, can contribute to poverty reduction and achieve food security (Plaza-Bonilla et al., 2015). Good agricultural practices, based on low tillage or no tillage, may result in the reduction of SOC breakdown and the enhancement of the soil carbon pool (Atallah et al., 2012; Cerdá et al., 2012; Boukhoudoud et al., 2016).

Quantifying SOC content in the NENA countries using available soil data is crucial, even at a small scale, to assess the nature and potential of available soil resources and analyze the associated threats. Mapping the spatial distributions of national and regional OC stocks can be used to monitor and model regional and global C cycles under different scenarios of soil degradation and climate change. Accurately quantifying SOC stocks in soils and monitoring their changes are considered essential to assessing the state of land degradation. At the same time, the predominantly calcareous soils of NENA region are rich in soil inorganic carbon (SIC). The dynamics of SIC and its potential in

sequestrating carbon in soils remain largely unknown and as such deserves thorough investigation. This paper analyzes the state of SOC and SIC in NENA countries and outlines challenges and barriers for devising organic carbon sequestration in NENA's impoverished and depleted soils. It also highlights several questions which scientists need to resolve. Finally, it discusses practical agricultural measures to promote SOC sequestration.

2. Materials and Methods

Data on SOC and soil inorganic carbon (SIC) contents in soils of the NENA region were retrieved from the soil database of the FAO-UNESCO digital soil map of the world (DSMW) at 1:5 Million. The database contains ~~large number of~~1700 georeferenced soil profiles collected and harmonized from each member state. These were excavated, sampled by horizon, down to the rock, and analyzed in the laboratory according to the standard world accepted methods (FAO, 2007). The soil map was prepared using the topographic map series of the American Geographical Society of New York, as a base, at a nominal scale of 1:5.000.000. Country boundaries were checked and adjusted using the FAO-UNESCO Soil Map of the World, on the basis of FAO and UN conventions. Soil classification was based on horizon designation, depth, texture, slope gradient and soil physico-chemical and chemical properties. Statistical (weighted) average was calculated for the topsoil (0-30 cm) and for the subsoil (30-100 cm) for the full series of chemical and physical parameters sufficient to assess main agricultural soil properties. To fill the gap in some attributes and complete the fields for which no data were available, an expert opinion internationally known soil scientists was used.

Using the DSMW and its updated attribute database maps of the SOC and SIC stock and distribution in 20 NENA states were produced. ~~The scale used in the DSMW is 1:5 Million~~ (FAO, 2007). ~~The soil map was prepared using the topographic map series of the American Geographical Society of New York, as a base, at a nominal scale of 1:5.000.000. Country boundaries were checked and adjusted using the FAO-UNESCO Soil Map of the World, on the basis of FAO and UN conventions.~~ To produce the maps representing the spatial distribution of SOC and SIC, ArcMap 10.3 was used to join the symbology of the C stocks and density with quantities classified into five numerical categories with natural breaks. The global LC maps at 300 m spatial resolution on an annual basis from 1992 to 2015 was produced by ESA. The Coordinate Reference System

The SOC content in studied soils varied between values as low as 0.13% and 0.16% and as high as 1.74% and 0.9% in topsoils and subsoils of Yermosols (Aridisols) and Rendzinas (Mollisols) respectively (Table 1). Worth noting that less than 20% of soil resources in NENA region have sequestered and accumulated SOC to an extend exceeding 1.0%. The highest SIC was observed in Solonchaks, Rendzinas and Aridisols that can be explained by the effect of the dominant calcareous rocks. The lowest SIC is detected in Lithosols and Xerosols subject to regular water and wind erosion removing the surface layer from eroded lands. The largest soil units are Yermosols (4670.6 Km$^2$), Lithosols (2914.3 Km$^2$) and Regosols (1193.2 Km$^2$). The first two soil classes and Xerosols (498.5 Km$^2$) are low resilient resistant to erosion and degradation. The most vulnerable to degradation soils are Solonchacks, Solonetz and Arenosols. Camboisosl, Fluvisols and Regosols possess high resilien resistance to erosion.

Table 1. Soil organic carbonOC and soil Sinorganic ICcarbon level in the major soil units of Near East North Africa region*

| Soil Type | Area, 1000 Km$^2$ | ResilienceResistance to Land Degradation | SOC content, % | | SIC content, % | |
|---|---|---|---|---|---|---|
| | | | topsoil | subsoil | topsoil | subsoil |
| Cambisols | 178.9 | Highly resilient | 0.90 | 0.48 | 0.25 | 0.64 |
| Fluvisols | 232.7 | Highly resilient | 0.65 | 0.24 | 1.12 | 1.40 |
| Kastanozems | 26.0 | Highly resilient | 1.50 | 1.00 | 1.69 | 3.96 |
| Regosols | 1193.2 | Highly resilient | 0.76 | 0.41 | 1.18 | 0.23 |
| Luvisols | 121.6 | Moderately resilient | 0.63 | 0.35 | 0.02 | 0.11 |
| Phaeozems | 3.8 | Moderately resilient | 1.46 | 0.63 | 0.40 | 0.70 |
| Rendzinas | 25.6 | Low resilience | 1.74 | 0.90 | 2.80 | 4.80 |
| Lithosols | 2914.3 | Low resilience | 0.97 | 0.40 | 0.01 | 0.06 |
| Vertisols | 45.4 | Low resilience | 0.69 | 0.52 | 0.45 | 0.72 |
| Xerosols | 498.5 | Low resilience | 0.36 | 0.25 | 0.25 | 0.45 |
| Yermasols (Aridisols) | 4670.6 | Low resilience | 0.13 | 0.16 | 2.50 | 2.30 |
| Solonchaks | 230.1 | Very Low low resilience | 0.49 | 0.36 | 3.60 | 3.90 |
| Solonetz | 31.2 | Very Low low resilience | 0.65 | 0.48 | 0.06 | 0.36 |
| Arenosols | 384.0 | Very Low low resilience | 0.87 | 0.10 | 0.00 | 0.00 |

*Source: DSMW, FAO, 2007

To assess the potential soil productivity in the NENA region, a land capability
model proposed by USDA (1999), which includes the soil geomorphological features
(geology and topography), other soil physic-chemical parameters conditioning soil fertility
like soil depth, texture, organic matter content, salinity and sodicity hazards was adopted.
The soils of the area were classified into four classes of arable soils: class I (highly
productive), class II (medium productive), class III (low productivity) and class IV (very
low productivity) and one non-arable soil class V, where lands suitable for wild vegetation
and recreation and lands with rock outcrops were grouped.

Arc Map 10.1 was used for the mapping of soil types and OC stock and density of
each soil unit based on the geographic or spatial distribution of the soil type. Total SOC
stocks and the stock of SOC were calculated separately for the topsoil (0-0.3m) and
subsoil (0.3-1.0m) using the following equations:

Total OC Stock (ton) = [Area (m$^2$)*Depth (m)*Bulk Density (ton m$^3$)*OC content (%)]/100
equation 1

Stock SOC (ton ha$^{-1}$) = Stock in~~of~~ given soil unit (ton)/Soil Unit Area (ha)    equation 2

The stocks of SOC under different land cover/land use were evaluated, as well.
Since 1990, the European Space Agency, (Climate Change Initiative project), started to
produce land cover (LC) maps of the NENA region. The version used in the study
(Website 1) corresponds to the second phase of the 2015 global LC. These maps have
300m of spatial resolution, using the Coordinate Reference System (CRS) in a geographic
coordinate system (GCS) based on the World Geodetic System 84 (WGS84) reference
ellipsoid. The legend assigned to the global LC map has been defined using the UN-
LCCS.
3. Results and Discussion
3.1. Land capability and SOC in NENA region
The most abundant soil classes in the NENA region are Arenosols, Xerosols and
Aridisols representing together more than 80% of available soil resources (FAO, 2007).
All three soil classes have low ~~resilience~~ resistance to degradation (Table 1). According to
the model of land capability, showing the proportion (%) of different soil ~~,~~productivity
classes in each country of NENA region, the majority of soils (40 to 100%) belongs to the
low, very low and non-arable classes (Figure 1).

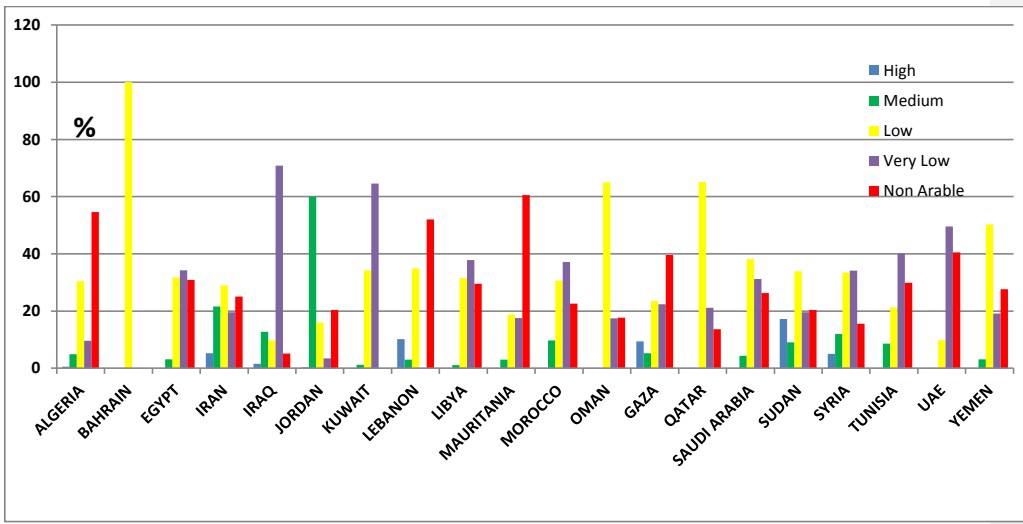

Figure 1. Land capability classification for the countries of NENA region, based on USDA Model (1999) and DSMW, FAO, 2007.

Thus, the proportion of highly and medium productive soils varies between 0% (Bahrain, Qatar, Oman and UAE) and 60% (Jordan). Countries like Iraq, Lebanon, Morocco, Palestine, Somalia, Syria and Tunisia present between 9 and 20% of highly to medium productive soils. The rest of NENA countries have less than 5% of their lands as high and medium productive soils. Some of these countries belong to the seriously endangered and food unsecure nations.

The potential medium productivity concept is based strictly on soil properties. But, with lack of water in drylands and prevalence of rainfed agriculture, the soil cannot show its full potential for food production. Similarly, irrigation with brackish and saline water restricts crop productivity due to the development of secondary soil salinity. When properly irrigated, the medium productive lands can provide moderately good harvests. For instance, our field observation in Jordan showed that due to climate change and climate variability, a large area of good lands was cropped with barley not because of land suitability but due to low rainfall (<200mm). In drought affected years, the land is converted into grazing area for small ruminants to make the minimal profit from the exploitation. The presence of kaolinite in red soils of Jordan developed from hard limestone under semi-arid climate points to the inheritance of material formed under more aggressive climate (Kusus and Ryan, 1985). The same was confirmed by Lucke et al.,

2013 for Red Mediterranean Soils of Jordan, which require new insights in their origin, genesis and role as a source of information on paleoenvironment.

The low productivity of the soil is reflected in the SOC contents. Two out of the three predominant soil classes (Xerosols and Aridisols) have SOC contents below 0.5% (Table 1). Overall, the NENA soils are poor in SOC, as less than 20% of soil resources have SOC contents exceeding 1.0%. The accumulation of SOC in NENA region is refrained by the high mineralization rate (Bosco et al., 2012). Climate change and recurrent drought events affect SOC sequestration in the soil. It is estimated that a rise in temperature of 3 $^o$C would increase the emission of carbon dioxide by 8% (Sharma et al., 2012). Among the soil properties affecting SOC, the clay and calcium carbonate contents are most relevant. Clay fraction tends to counteract the decomposition of SOC, as found in clay soils of Morocco and in Vertisols of northern Syria (FAO and ITPS, 2015). But the dominant soil classes, Xerosols, Aridisols or Arenosols (Table 1) characterized by sandy and sandy loam textures, are subject to fast decomposition.

Next to the clay texture, the presence of calcium carbonate decreased the decomposition of composted organic material in sub-humid coastal Lebanon (Al Chami et al., 2016). This slower turnover of organic matter was explained by the low porosity and prevalence of micropores in soil macro-aggregates (Fernãndez-Ugalde et al., 2014). For the SIC, the highest values are found in soil classes dominated by calcareous rocks, that is the Solonchaks, Rendzinas and Aridisols (Table 1). The lowest stocks were detected in Lithosols and Xerosols, subject to water and wind erosions that remove the surface layer of eroded lands.
.

3.2. Mapping of soil carbon stocks in the soils of NENA region

The choice of scale when using or producing soil maps to show the spatial distribution and estimate national or regional C stock and density may lead to uncertainty in small countries and fragmented landuse (Darwish et al. 2009). In the absence of more detailed, accessible, regional and national soil databases, the use of small scale maps is justified. Only in December 2017, the GSP-FAO, ITPS launched the version 0.1.0 of the global soil organic carbon map, showing the SOC stock in topsoil (http://www.fao.org/3/a-i8195e.pdf). An overview, reconnaissance carbon stock and density mapping for NENA region in topsoil and subsoil is justified to preliminary assess regional SOC stock both in

topsoil and subsoil and compare and analyze the common problems and challenges in C sequestration using unified background information provided by member states and harmonized in the FAO-UNESCO DSMW. The recent attempt undertaken in this paper was done despite the uncertainty associated with the scale of mapping where small scale maps join polygons having area below the smallest mappable unit, considered equivalent to or less than 0.5 cm x 0.5 cm area, i.e., $<0.25cm^2$, with the neighboring mappable soil polygons. Thus, the need for more detailed and harmonized soil mapping and coding of available national information arises to downscale to national and local soil assessment and mapping, which is currently on the agenda of the Global Soil Partnership (GSP).

The majority of the countries of NENA region presents moderate to relatively low total stocks of SOC. This is especially relevant to the Gulf countries, Iran, Tunisia and Morocco, with values below 221 Mega tons (Figure 2). Such low OC sequestration potential can be explained by the prevalence of arid climate and rare natural vegetation and the reliance on irrigation to produce food and feed crops. A regional implementation plan for sustainable management of NENA soils appeared in 2017 (Website 2). In fact, the total stocks, in the topsoils (0-0.3 m) of NENA countries, represent only 4.1% of the world global stock (Website 3).

The stock of SOC (ton ha$^{-1}$) was mapped as well (Figure 3). A large proportion (69%) of the soil resources presents a stock inferior to 30 tons ha$^{-1}$ (Figure 3), value considered as a threshold (Batjes and Sombroek, 1997). This could be linked to the relief of these countries, with rare mountainous landscapes that enjoy a more humid climate and a longer duration of soil moisture.

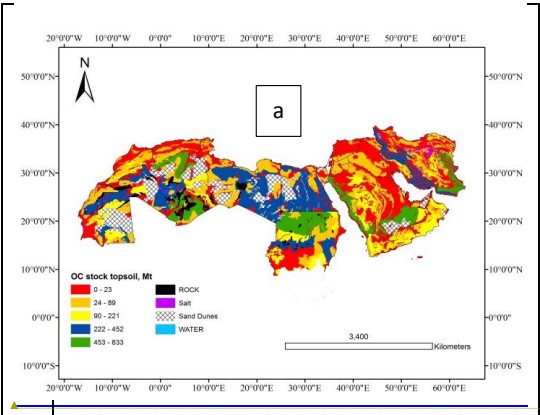 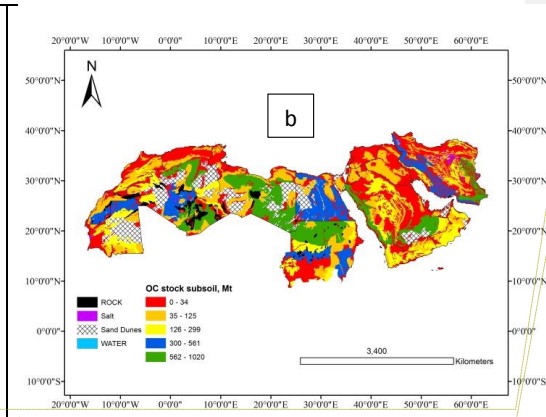

Figure 2. Spatial view of total soil organic carbon stock across the countries of NENA region, Mega ton (a-toposil; b-subsoil).


257        Mapping was done on a small scale, which could be a source of a loss of

information. To test this, the results of the current estimation (1:5 Million) of SOC stocks
in Lebanon, were compared with the large scale mapping undertaken recently to produce
the unified soil map of Lebanon at 1:50,000 (Darwish et al.;, 2006 1:50,000). This
comparison showed discrepancies between 11% for the topsoil and 14% for the subsoil
(Darwish and Fadel, 2017). Therefore, the level of uncertainty falls within the admitted
diagnostic power of soil mapping, estimated to be close enough to the reported range of
map units' purity in reference areas, i.e., a matching between 65% and 70% (Finke et al.,
2000). Loss of information related to small, non-mappable soil units in small scale
mapping (1:5 Million or 1:1 Million) could be corrected by national and subregional large
scale soil mapping (1:50,000 and 1:20,000).

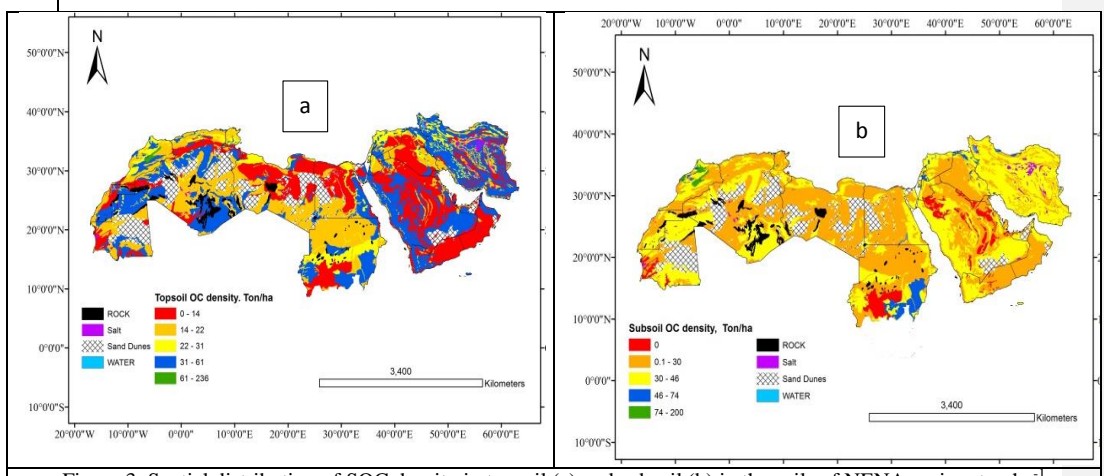

Figure 3. Spatial distribution of SOC density in topsoil (a) and subsoil (b) in the soils of NENA region, ton ha[-1]


3.3. Land cover mapping and effect on SOC stock

270        Land cover map of NENA region shows nearly 80% of the area is covered with

bare lands (Figure 4a, b). Grassland, sparse vegetation cover and rainfed agriculture are
close by area varying between 4.39% and 5.27%. The irrigated crops do not exceed 0.66%
of the total area. Apparently for this reason, The region is becoming increasingly
dependent on food imports, because of demographic pressure, rapid urbanization, water
scarcity and climate change (FAO, 2015).

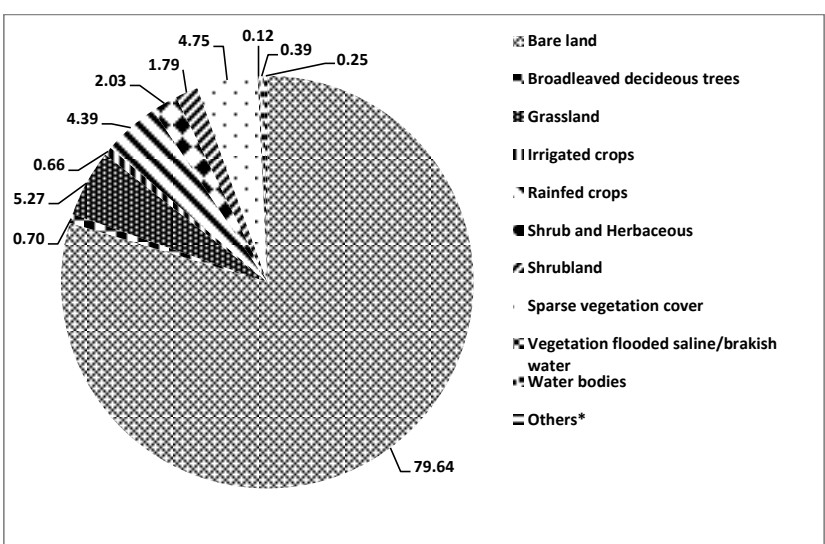

*Others: Mixed trees, Neadleaved evergreen trees and Urban

Figure 4a. Proportion of main land cover and land use in NENA region (Source: ESA, 2015)

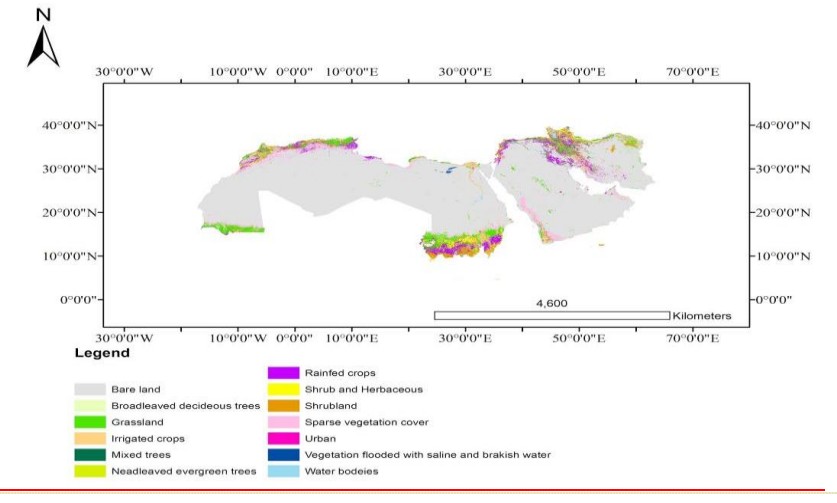

Figure 4b. Land cover map of NENA region (Source: ESA, 2015; http://maps.elie.ucl.ac.be/CCI/viewer/index.php)

Comparing our results with the global SOC map produced by Hengl et al., (2014), based on soilGrids1km layers showing the soil organic carbon content in permille in 0-5 cm and the predicted global distribution of the soil organic carbon stock in tonnes per ha for 0–200 cm to be beyond the followed by FAO methodology of SOC stock estimation and presentation. In this paper the standard methodology of the measured SOC stock and density in topsoil (0-30 cm) and subsoil (30-100 cm) was followed. The first Global SOC Map was launched on December 5, 2017. However, a comparison of values of SOC

content (%) and SOC stock revealed comparable trends values for the C content and stock (1-2% and 20-204 ton/ha), with higher upper density in Hengel et al approach.

FurtherIn our study, the combination of SOC stock map with the land cover map showed, the significant effect of land cover on SOC stocks in NENA region. of SOC were studied in relation to land cover/land use. As can be expected, Shrublands, sparse vegetation and bare lands gave the smallest values, between 14 and 26 ton ha$^{-1}$ (Figure 45). In a mixture of shrublands and herbaceous vegetation, the SOC increases to 40 ton ha$^{-1}$. The highest density (30 and 60 ton ha$^{-1}$) is found under forest stands.

Despite the expected impact of frequent plowing, the soils under mixed trees and irrigated crops have higher density than rainfed crops. The highest SOC stock was observed under evergreen forest land whose area is very limited (3380 km$^2$ corresponding to 0.02% from the total area). Surprisingly, the stock found under urban soils ($\approx$ 30 tons ha$^{-1}$) was moderate. This could be related to the urban encroachment on prime soils. Between 1995 and 2015, rapid urban growth caused the loss of over 53 Million tons of soils, 16% of which correspond to prime soils (Darwish and Fadel, 2017). The assessment of SOC content in time and space in relation to land cover showed a decline of OC content in topsoil by up to 1% between 2001 and 2009 (Stockmann et al., 2015). Land cover change was considered as the primary agent that influences SOC change overtime, followed by temperature and precipitation.

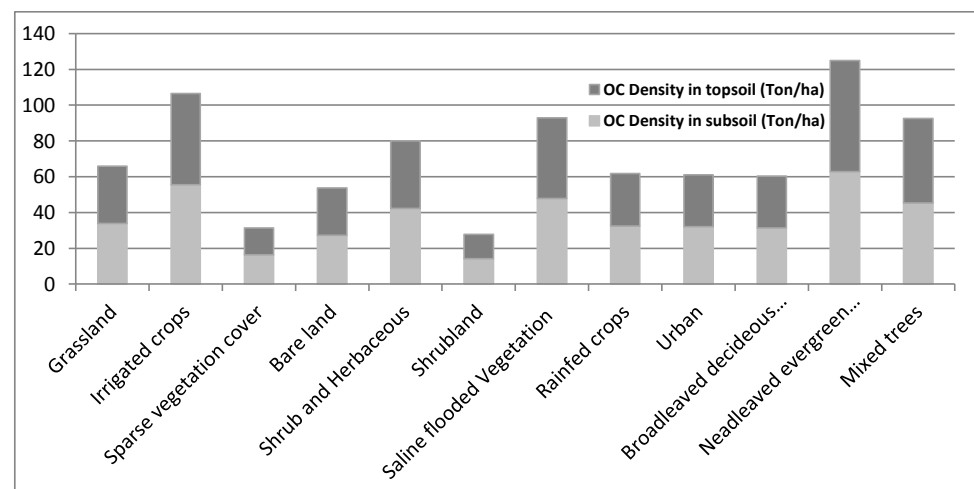

Figure 4~~5~~. SOC ~~stock~~ density (tons ha$^{-1}$) in the topsoils (0-0.3m) and subsoils (0.3-1.0m),
calculated from the FAO DSMW (FAO, 2007) on corresponding land cover
( http://maps.elie.ucl.ac.be/CCI/viewer/index.php).~~as related~~
In addition to the stocks of SOC in relation to land cover/land use, the stocks of
SOC and SIC were established per country (Figure 6~~5~~). The stock of SIC was compared to
that of SOC (Figure 5). The range of SIC stocks is very wide, from less than 25 tons ha$^{-1}$
(Gaza subsoil) to 450 tons ha$^{-1}$ (Bahrein subsoil), while that of SOC varied between ≈ 20
tons ha$^{-1}$ (Bahrein subsoil) and 45 tons ha$^{-1}$ (Sudan subsoil). Based on the stocks of SIC in
the subsoils, the countries were separated into three groups. The first, represented by six
countries (Bahrein, Oman, Egypt, Saudi Arabia, UAE and Yemen) was dominated by
calcareous parent materials, with values in the subsoil exceeding 200 tons ha$^{-1}$ (Figure 5).
The second group, with eight countries (Kuwait, Libya, Iran, Iraq, Algeria, Qatar,
Morocco and Tunisia), presents a SIC density between 100 and 200 tons ha$^{-1}$. Finally, the
third group (Gaza, Jordan, Lebanon, Mauritania, Syria and Sudan) has less than 100 tons
ha$^{-1}$.

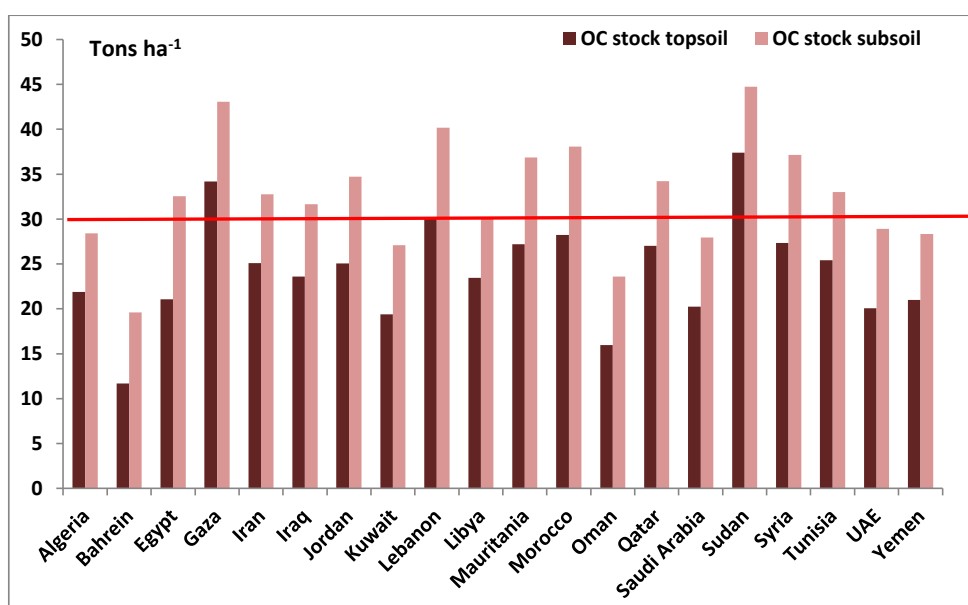


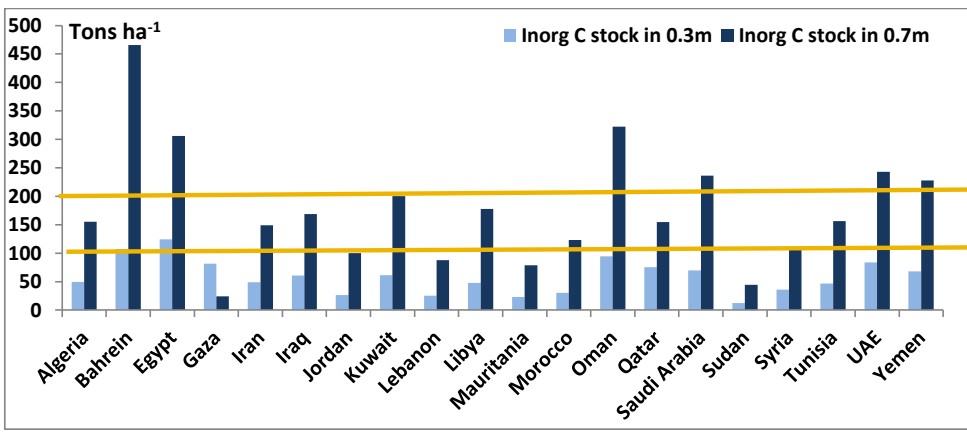

Figure 65. Stocks of soil organic carbon and soil inorganic carbon (tons ha⁻¹) in the topsoils (0-0.3 m) and subsoils (0.3-1.0 m) of the 20 countries in the NENA region. The red line represents the threshold for organic carbon, and the yellow lines correspond to the limits of classes.

3.3. Challenges of carbon sequestration in NENA agroecosystems

Climatic conditions characterized by wetting/drying cycles, a long dry and hot season (Boukhoudoud et al., 2016) promote the decomposition of SOC. Further, frequent cultivation, irrigation with saline water, and soil salinity rise oin coastal areas the prevailing agriculture practices (Boukhoudoud et al., 2017) exert significant effects on soil microbial functional properties (Boukhoudoud et al., 2017). In this section, there will be a presentation of major practices affecting SOC followed by a discussion of preventive and remediation measures.

3.3.1. Tillage and SOC

Tillage practices contribute to the vulnerability of soils to water erosion. If not properly managed, some 41 million hectares would be affected by water erosion (FAO and ITPS, 2015). The erosion of soil surface layers can affect the soil carbon in two possible ways. The greater exposure of carbonates to climatic elements could increase the loss of SIC to the atmosphere and ground water. Compared to stable soils,Also, the higher decomposition of SOC in eroded soils decreases the productivity of cultivated crops and can reduce SOC stock if not properly managed (Plaza-Bonilla et al., 2015).

A possible measure to reduce the risk of erosion is the no-tillage. No-tillage coupled with mulching, to reduce weed development and omit herbicide asapplication, as part of conservation agriculture (CA), aims to keep more plant residues on soil surface,

enhance C sequestration, increase soil aggregates, improve water infiltration and to provide a protection to soil carbon from decomposers (Palm et al., 2014). Through a modification of common practices, such as the frequency and depth of tillage, changes in the SOC could be promoted in most soils. Experiments conducted by ICARDA, Syria, showed that no-tillage performed well in terms of energy and soil conservation (Plaza-Bonilla et al., 2015). Elsewhere, in Palestine soil conservation was found to pay, with a net profit 3.5 to 6 times higher than without conservation measures (FAO and ITPS, 2015). In dryland regions, agricultural activities based on CA practices are beneficial as crop residues are left on the soil surface (Plaza-Bonilla et al., 2015). The presence of residues would protect the soils from high evaporation, water and wind erosions. This is especially relevant to soils that are sensitive to degradation, such as the very shallow Lithosols, the easily periodically wetted (swelling) and dry (shrinking) Vertisols, Gypsic Yermasols (Aridisols), the poorly-structured Solonchaks and Solonetz, the sandy-textured Arenosols, and the desert soils (Xerosols).

Major constraints facing soil conservation measures, in East Mediterranean, were due to knowledge and perception, prevailing practice of complete removal and some times burning of residues after harvest, land tenure and type of landscape (FAO, 2012; FAO and ITPS, 2015). These major factors are socio-economic in nature, rather than scientific. They are related to the ability of growers to accept new techniques and adopt them. In many situations, the transfer from the research stations to the farmers was not smooth. For instance, CA was successfully tested in experimental stations in Morocco and Lebanon, but several social and technical barriers prevented it from reaching farmers (Mrabet et al., 2012; FAO, 2012).

A debate has been taking place about the effect of no-tillage on SOC. Most Many authors agree that under CA, SOC increases near the soil surface, but not necessarily throughout the profile. A study compared 100 pairs, where no-tillage has been practiced for over 5 years. The absence of tillage lead to higher C stocks (0-30 cm soil depth) in 54% of pairs, while 39% showed no difference in stocks (Palm et al., 2014). In the absence of tillage, the slower decomposition of residues would result in higher belowground C accumulation on the soil surface. Over a period of 5 years, zero tillage promoted an increase in SOC equal to 1.38 Mg ha$^{-1}$ as compared to the conventional tillage in northern Syria (Sommer et al., 2014).

392

### 3.3.2 Agricultural practices and SOC

Practices, such as the application of N fertilizers, ~~of~~ the organic amendments, the incorporation of residues and crop rotations, influence the levels of SOC. The lack of accessible nutrients and soil mining make most crops entirely reliant on accumulated SOC (Plaza-Bonilla et al., 2015). In East Africa, 14-years of continuous cultivation without any inputs, decreased SOC from 2% to 1% (Sharma et al., 2012). The application of N fertilizers was associated with increased levels of soil C, as compared to the absence of N fertilizers (Palm et al., 2014). In a 10-year rotation of wheat-~~grain~~ legume (vetch) in northern Syria, the application of nitrogen fertilizers to the cereal caused a notable increase of SOC, in the top 1m of soil, equal to 0.29 Mg ha$^{-1}$ year$^{-1}$ (Sommer et al., 2014). Similarly, the growth of intercropped legumes as winter cover ~~cropcrop legumes~~ (Vicia sp., Lathyrus sp.~~s~~) alone or with barley (Hordeum vulgaris), between cherry trees in semi-arid Lebanese area (Jourd Aarsal, eastern Lebanese mountains), ~~—~~increased SOC significantly notably when legumes were mixed with barley (Darwish et al., 2012). Results showed that the sites were supplemented with OM varying between 140 and 250 kg ha$^{-1}$season$^{-1}$ resulting from the decomposition of plant root residues. The above ground plants provided the orchards with 95-665.7 kg ha$^{-1}$season$^{-1}$ of OM. Plant residues provided additional feedstuff for small ruminants; the soils were enriched with OM and fixed nitrogen with more efficient use of surface soil moisture.

The effects of crop rotations on SOC are related to the amounts of above and belowground biomass produced and retained in the system. In a study conducted in semi-arid northern Syria, a 12-year rotation gave higher SOC in wheat-medic (12.5 g SOC kg$^{-1}$ soil) and wheat-vetch (13.8 g SOC kg$^{-1}$ soil) rotations, as compared to continuous wheat (10.9 g SOC kg$^{-1}$ soil) or wheat-fallow (Masri and Ryan, 2006). In this rainfed system, the introduction of a forage legume (vetch/medic) with wheat, over a decade, was able to significantly raise the level of SOC. Further, the combination of crop rotations and no-tillage was found to sequester more C than monocultures (Palm et al., 2014). One means of building up biomass is through cover winter crops. Their beneficial impact on C sequestration and water infiltration has been demonstrated. The presence of a cover on the soil surface protects the soil against erosion. In the NENA region, their cultivation is restricted to sub-humid to humid areas (> 600 mm of rainfall). Still, more research is

needed about the best species to be used, the optimum termination strategies of the cover
crop as well as the best date (Plaza-Bonilla et al., 2015).
In poor dryland regions Poverty, especially in the rainfed agricultural systems,
prevents some practices leads to the removal of all such as the incorporation of residues.
Overall, crop residues serve as fodder or for household cooking or heating, leaving little
remains ion the soil surface. Even animal dung is used as cooking fuel in many regions.
The low SOC content could be improved by increasing the crop residues produced and
incorporated.  Such an approach requires the application of fertilizers in order to avoid the
depletion of soil nutrients (Plaza-Bonilla et al., 2015).
Some authors question the validity of remediation measures to build-up SOC in
most of the NENA region. Results from research stations in Egypt and Syria provide
evidences to the contrary. In a trial in north-east Cairo, Egypt, the irrigation of a sandy soil
with sewage water, for 40 years, changed its texture to loamy sand (Abd el-Naim et al.,
1987).  This modification of the soil texture leads to a significant improvement of the soil
physical properties. Further, within the same long-term trial, the irrigation with sewage
water, for 47 years, increased SOC to 2.79%, against 0.26% in the control (Pescod and
Arar, 2013).  This rather slow accumulation could be related to the sandy soil texture and
to the input of the organic matter in labile, soluble forms. The addition of more stable
composted materials was tested in semi-arid north Syria. The amount of compost, 10 Mg
ha$^{-1}$ every two years, needed to raise the SOC, was too large in these rainfed systems.
Rather than relying on composts, the authors found that a combination of reduced tillage
and a partial retention of crop residues moderately increased SOC (Sommer et al., 2014).
The quality of residues seems to affect the SOC on the short-term but on the medium-term
it is the quantity that matters (Palm et al., 2014).
3.3.3. Impact of irrigation on agricultural soils
The irrigated land might represent a minor fraction of agriculture in NENA region,
but irrigated crops are essentially found on prime soils (Figure 4). Frequent wetting of
irrigated soils make them more likely to lose C as compared to dry soils.  Lack of moisture
limits soil mineralization (Sharma et al., 2012). Irrigated soils promote intense microbial
activity and a rapid decomposition of SOC. In the fertile region of Doukkala, Morocco,

known for producing wheat and sugar beet, a decade of irrigated farming decreased SOM by 0.09% per year (FAO and ITPS, 2015). This loss could have been reduced through the incorporation of crop residues. But, in these mixed farming systems, residues are consumed by farm animals.

The irrigation of soils in NENA region is expected to affect the SIC. Dryland soils were considered to contain, at least, as much SIC as SOC (Sharma et al., 2012). However, this study showed much higher SIC than SOC, notably in the subsoils. Despite this large stock, there is a major knowledge gap regarding the effects of land use and management on the dynamics of SIC. This is especially relevant to the irrigation with calcium or sodium-enriched groundwater (Plaza-Bonilla et al., 2015). In these conditions, the formation of calcium carbonate could be accompanied by some release of carbon dioxide while the development of sodicity can cause irreversible SOC loss.

4. Conclusions

~~The~~ NENA area consisting of 14% of the earth surface area contributes only 4.1.% of total SOC stocks ~~of~~ in ~~SOC (0-0.3m)~~ topsoil ~~showed a small contribution to global SOC stock in the topsoil (4.1%), against 14% of the earth surface area~~. The ~~s~~Soil resources ~~of s of the~~ NENA region are developed under dry conditions with prevailing of rainfed agriculture. ~~mostly highly vulnerable to~~Achieving land degradation neutrality ~~degradation,~~ and food security ~~will~~ depends much on land stewardship and sustainable ~~agricultural measures~~management of soil resources. The land capability model showed that most NENA countries (17 out of 20), suffer from low productive lands (>80%). ~~To obtain an idea of the status of the soil carbons, the spatial distributions of SOC and SIC stocks for the NENA region were mapped (1:5 Million). This small scale mapping was compared with a larger scale mapping (1:50,000) for Lebanon. A moderate discrepancy (11% to 14%) was found between the two scales.~~ The ~~results of the mapping Mapping~~current mapping of ~~the stocks of~~ SOC and SIC density showed that 69% of soil resources present a SOC stock ~~of SOC~~ below the threshold of 30 tons ha$^{-1}$. The ~~stocks~~ density varied between ≈ 10 tons ha$^{-1}$ in shrublands and 60 tons ha$^{-1}$ for evergreen forests. Highest stocks were found in forests, irrigated crops, mixed orchards and saline flooded vegetation. The moderate ~~stock~~ density (≈30 tons ha$^{-1}$) in urban areas indicates that some urban growth was at the expenses of prime soils. The stocks ~~-~~of SIC were higher than ~~those of~~ SOC

density, indicating the calcareous nature of soils. In subsoils, the SIC ranged between 25 and 450 tons ha$^{-1}$, against 20 to 45 tons ha$^{-1}$ for SOC.

~~Decomposition of SOC is accelerated by climatic conditions, high temperatures, wetting/drying cycles, and by sandy soil textures.~~ Although OC sequestration in the NENA region is problematic, this task is still possible, requiring the protection of the topsoils and sustainable land management. Practices of conservation agriculture (no-tillage, intercropping and agro-pastoral system, ~~presence of~~winter soil cover, proper rotation…) could be effective as the presence of residues reduces the evaporation, as well as water and wind erosions and promote the aboveground biomass production. This is especially relevant to soil classes that are susceptible to degradation. ~~Further, the combination of crop rotations and no-tillage was found to sequester more C than monocultures.~~ In semi-arid regions, t~~T~~he introduction of legumes, as part of a cereal-legume rotation, and the application of nitrogen fertilizers to the cereal caused a notable increase of SOC, after 10 years. ~~The effects of crop rotations on SOC are related to the amounts of above and belowground biomass produced and retained in the system.~~ A faster result was achieved through winter cover crop consisting of fruit trees-legume-barley intercropping system.

~~Some k~~Knowledge gaps exist, especially in aspects related to the effect of irrigation on SOC, as well as on SIC ~~at the level of soil profile and soil landscape~~. Still, major constraints facing soil conservation measures and carbon sequestration are socio-economic in nature, rather than scientific. They are related to the ability of growers to accept new techniques and adopt them. Awareness raising and capacity building at the level of stakeholders and decision--makers in the NENA region can contribute to alleviate the pressure on ~~vulnerable~~ soil resources, improve SOC sequestration~~,~~ and maintain soil ~~resilience~~resistance to degradation ~~and strengthen food security.~~

5. Acknowledgements

This paper was supported by the FAO, GSP-ITPS, UNESCWA and CNRS Lebanon within a land degradation assessment project.

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

FAO, 2012. Country Study on Status of Land Tenure, Planning and Management in
Oriental Near East Countries Case of Lebanon. FAO, RNE, SNO. Cairo, Egypt:
161p.
FAO, 2015. Regional Overview of Food Insecurity - Near East and North Africa:
Strengthening Regional Collaboration to Build Resilience for Food Security and
Nutrition, Cairo, Egypt, FAO.
FAO, 2017. Voluntary Guidelines for Sustainable Soil Management Food and Agriculture
Organization of the United Nations Rome, Italy
FAO and ITPS, 2015. Status of the World's Soil Resources (SWSR) – Main Report. Food
and Agriculture Organization of the United Nations and Intergovernmental
Technical Panel on Soils, Rome, Italy.
Fernandez-Ugalde O., Virto I., Barre P., Apesteguia M., Enrique A., Imaz M.J. and P.
Bescansa (2014). Mechanisms of macroaggregate stabilisation by carbonates:
implications for organic matter protection in semi-arid calcareous soils. Soil
Research 52: 180-192.
Finke, P., Hartwich, R., Dudal R. Ibanez J. Jamagne M. King D. Montanarella L and N.
Yassoglou (2000) Georeferenced Soil Database for Europe. Manual of Procedures,
Version 1.1. European Soil Bureau.
Guo, LJ., Lin, S., Liu, TQ., Cao, CG., Li, CF. (2016) Effects of conservation tillage on
topsoil microbial metabolic characteristics and organic carbon within aggregates
under a rice (Oryza sativa L.)-wheat (Triticum aestivum L.) cropping system in
Central China. PLoS One 11:e0146145.
Hengl, T., de Jesus, JM., MacMillan, RA., Batjes, NH., Heuvelink, GBM., Ribeiro, E. ,
Rosa, AS. , Kempen, B., Leenaars, JGB., Walsh, MG., Gonzalez, MR. (2014)
SoilGrids1km-Global Soil Information Based on Automated Mapping. PLoS ONE
9(8): e105992. doi:10.1371/journal.pone.0105992.
Lal, R. (2003) Soil erosion and the global carbon budget. Environment International, 29
606      (4): 437-450

Masri, Z. and J. Ryan (2006) Soil organic matter and related physical properties in a
Mediterranean wheat-based rotation trial. Soil Tillage Research 87: 146–154.
Mrabet, R., Moussadek, R., Fadlaoui, A., and E. van Ranst (2012) Conservation
agriculture in dry areas of Morocco. Field Crops Research 132: 84-94.

Pescod, M.B. and  A. Arar (2013)  Treatment and use of sewage effluent for irrigation. Butterworths. https://books.google.com.lb/books. Isbn: 1483162257. Pages 211-212

Plaza-Bonilla, D., Arrúe, JL.,  Cantero-Martínez, C.,  Fanlo, R.,  Iglesias, A., and Álvaro-Fuentes J.  (2015) Carbon management in dryland agricultural systems. A review. Agronomy  for Sustainable Development 35 (4): 1319-1334.

Sharma, P., Abrol, V., Abrol, S., and R. Kumar (2012) Climate change and carbon sequestration in dryland soils. In Tech open access book, chapter 6. 26 pages. http://dx.doi.org/10.5772/52103

Sommer, R., Piggin, C., Feindel, D., Ansar, M., van Delden, L., Shimonaka, K., Abdalla, J., Douba, O., Estefan, G.,. Haddad, A., Haj-Abdo, R., Hajdibo, A., Hayek, P., Khalil, Y., Khoder, A., and J. Ryan (2014) Effects of zero tillage and residue retention on soil quality in the Mediterranean region of northern Syria. Open Journal of Soil Science 4 (3), Article ID: 44383.  17 pages. DOI:10.4236/ojss.2014.43015

Stockmann, U., Padarian, J., McBratney, A., Minasny, B., deBrogniez, D., Montanarella , L., Young Hong, S., Rawlins, BG., Damien, J. (2015) Field Global soil organic carbon assessment. Global Food Security (6): 9-16.

USDA (1999). Land capability classification. NRCS-USDA.

von Grebmer, K.,  Bernstein, J., Hossain, N. , Brown, T., Prasai, N., Yohannes, Y., Patterson, F., Sonntag, A., Zimmermann, S.M., Towey, O., and C Foley (2017) Global Hunger Index: The Inequalities of Hunger. Washington, DC: International Food Policy Research Institute; Bonn: Welthungerhilfe; and Dublin: Concern Worldwide.

Yu, L., Dang, Z-Q., Tian, F-P., Wang, D., and G-L. Wu (2017). Soil organic carbon and inorganic carbon accumulation along a 30-year grassland restoration chronosequence in semi-arid regions (China). Land Degrad. Develop., 28: 189–198.

Webography
Website 1. http://maps.elie.ucl.ac.be/CCI/viewer/index.php
Website 2. http://www.fao.org/3/a-bl105e.pdf
Website 3. https://drive.google.com/drive/folders/1454FhX_p2_GxjZT-fsi0-RncL2QpSN