# Peer review of "Challenges of soil carbon sequestration in NENA Region"

_SOIL, 2017_

## Referee Comment (RC1) · Anonymous Referee #1 · 6 Mar 2018

General comment: The manuscript analyzes the SOC/SIC content in the NENA region and present some options to improve SOC, which are useful for guiding decision making process, particularly at regional and maybe national level. however, the conclusion that the potential for carbon sequestration in NENA region is low needs some careful consideration: (a) due to the vast geographic area the NENA region is extended over, which make any slight potential per hectare is significant at regional level, and (b) because there are no scenarios presented to check what would be the potential if some sustainable soil/land management practices are introduced in the region. therefore, a careful consideration should be given to this conclusion, which might also influence important decisions to be made as sustainable soil and land management are considered within the climate change context.

[Figure]

Specific Technical comments: The term resilience is used in various contexts throught out the manuscript (resilience to erosion, resilience to degradation), please check if the proper term would be "resistance".

Specific corrections In the abstract: the first word should be "Near" not "North" please spell out SIC please elaborate little on the socio-economic constraints

page 5: please check 60% of Jordan is medium productive is very high figure given that 90% of the country receives less than 200mm of annual rainfall.

page 11, second paragraph, need to link conservation agriculture with residue management, which is elaborated later but need to be emphasized here as well.

---

## Referee Comment (RC2) · Anonymous Referee #2 · 6 Mar 2018

This paper addresses a very important topic for soil degradation and food security. This topic could potentially interest a wide audience readership and is well suited to the journal 'SOIL'.

However I have several concerns about the paper, which I explain below.

I was rather disappointed by the method used by the authors, which is basically populating large soil map units with mean values taken from profiles. I'm not sure at all it is relevant and new to do so. I would have expected that at least climate data and land use data are added to the dominant soil type information to do this exercise. I know of course that from a very global point of view, soil types and climate are related but this is not always the case (e.g. you can find rendzinas, fluviosols, cambisols, lithosols, arenosols, etc. in nearly all parts of the World). I'm also disappointed by the fact that

[Figure]

the authors do not integrate land-use in their mapping, though they show it has a major effect. This seems to me contradictory.

I would have expected a more novel approach in mapping such as the use of Digital Soil Mapping. By the way, there have several attempts to map SOC at the global scale and the authors seem to ignore them (e.g. maps from Stockmann et al; Hengl et al; and the recent exercise under the umbrella of the FAO). I would at least compare the results from this study with previous ones in a discussion part.

There is quite no discussion about uncertainty, no error bars or box-plots in results, and this is a serious concern. Validation against Lebanon may induce a serious bias as Lebanon soils have been much more known and investigated for a very long time than soils from some other countries. Subsequently it may be that the DSMW is much more precise in Lebanon than in other regions.

The Mat&Meth section is not enough detailed. We don't know how many profiles were used (a 'large number'), how many per soil type, if they were georeferenced or not and, if so, with which accuracy, when these profiles were taken (they might be no more representative of SOC stocks if they were taken 50 years ago).

We also do not know how the calculations by land use were done. By a geographical way, or by extracting soil use classes from the DSMW? There is also here a question of matching observations both in space and time.

Finally the general discussion about the challenges is rather interesting but quite fully disconnected from the results.

More detailed comments

The legend of Fig 2 is unclear (Mega ton on which area?)

The threshold of 30 tons is highly questionable

There is twice the same sentence lines 90-92 and lines 101-103.

[Figure]

There is a problem with the labelling of countries (Fig. 5 top)Abstract first word shoul be Near and not North

---

## Author Comment (AC1) · 6 Mar 2018

Reply to Reviewer 1

1. Reply to the general comments . . . . . . ..however, the conclusion that the potential for carbon sequestration in NENA region is low needs some careful consideration: (a) due to the vast geographic area the NENA region is extended over, which make any slight potential per hectare is significant at regional level, and (b) because there are no scenarios presented to check what would be the potential if some sustainable soil/land management practices are introduced in the region. Therefore, a careful consideration should be given to this conclusion, which might also influence important decisions to be made as sustainable soil and land management are considered within the climate

change context.

We completely agree with the reviewer note about the sensitivity of the issue and that we shall create a positive attitude towards the efforts of international community to improve land stewardship and strengthen governance to improve C sequestration in NENA region despite the difficult situation in this part of the world, where the majority of lands represents deserted and rainfed areas. Therefore, the conclusion was changed into the following text with several scenarios included briefly here but elaborated in the results:

NENA area consisting of 14% of the earth surface area contributes only 4.1.% of total SOC stocks in -topsoil. The soil resources of NENA region are developed under dry conditions with prevailing of rainfed agriculture. Achieving land degradation neutrality and food security depends much on land stewardship and sustainable management of soil resources. The land capability model showed that most NENA countries (17 out of 20), suffer from low productive lands (>80%). The current mapping of SOC and SIC density showed that 69% of soil resources present a SOC stock below the threshold of 30 tons ha-1. The density varied between $\approx$ 10 tons ha-1 in shrublands and 60 tons ha-1 for evergreen forests. Highest stocks were found in forests, irrigated crops, mixed orchards and saline flooded vegetation. The moderate density ($\approx$30 tons ha-1) in urban areas indicates that some urban growth was at the expenses of prime soils. The stocks of SIC were higher than SOC density, indicating the calcareous nature of soils. In subsoils, the SIC ranged between 25 and 450 tons ha-1, against 20 to 45 tons ha-1 for SOC.

Although OC sequestration in the NENA region is problematic, this task is still possible, requiring the protection of the topsoils and sustainable land management. Practices of conservation agriculture (no-tillage, intercropping and agro-pastoral system, winter soil cover, proper rotation. . .) could be effective as the presence of residues reduces the evaporation, as well as water and wind erosions and promote the aboveground biomass production. This is especially relevant to soil classes that are susceptible

to degradation. In semi-arid regions, the introduction of legumes, as part of a cereal-legume rotation, and the application of nitrogen fertilizers to the cereal caused a notable increase of SOC, after 10 years. A faster result was achieved through winter cover crop consisting of fruit trees-legume-barley intercropping system.

Some scenarios include the following:

A. In a 10-year rotation of wheat- legume (vetch) in northern Syria, the application of nitrogen fertilizers to the cereal caused a notable increase of SOC, in the top 1m of soil, equal to 0.29 Mg ha-1 year-1 (Sommer et al., 2014).

B. Similarly, the growth of intercropped legumes as winter cover crop (Vicia sp., Lathyrus sp.) alone or with barley (Hordeum vulgaris), between cherry trees in semi-arid Lebanese area (Jourd Aarsal, eastern Lebanese mountains), increased SOC significantly notably when legumes were mixed with barley (Darwish et al., 2012). Results showed that the sites were supplemented with OM varying between 140 and 250 kg ha-1season-1 resulting from the decomposition of plant root residues. The above ground plants biomass provided the orchards with 95-665.7 kg ha-1season-1 of OM. Plant residues provided additional feedstuff for small ruminants; the soils were enriched with OM and fixed nitrogen with more efficient use of surface soil moisture.

Reply to specific comments -The term resilience is used in various contexts throught out the manuscript (resilience to erosion, resilience to degradation), please check if the proper term would be "resistance".

The term resilience was changed into resistance.

Specific corrections In the abstract: the first word should be "Near" not "North" please spell out SIC please elaborate little on the socio-economic constraints

Done

page 5: please check 60% of Jordan is medium productive is very high figure given that 90% of the country receives less than 200mm of annual rainfall.

In this paper potential soil productivity was modeled based on the USDA model (1999). The potential medium productivity is based strictly on soil properties. But, with lack of water in drylands and prevalence of rainfed agriculture, the soil cannot show its potential for food production. Similarly, irrigation with brackish and saline water restricts crop productivity. When properly irrigated, these lands can provide moderately good harvests. I visited Jordan several times and undertook field visits to classify soils and assess land degradation and saw vast good lands cropped with barley because of lack of irrigation water. Often, even barley fail in central and east Jordan (<50 mm rain) and land is immediately converted into grazing land for small ruminants. The soil studies done by John Ryan more than 45 years ago on the red soil of Jordan showed relict soils carrying properties of higher rainfall not relevant to actual climatic conditions. . .The same was confirmed by: Lucke, B., Kemnitz, H., Bäumler, R., Schmidt, M. (2013): Red Mediterranean Soils in Jordan: New insights in their origin, genesis, and role as environmental archives. ‐ Catena, 112, 4‐24 DOI: 10.1016/j.catena.2013.04.006

page 11, second paragraph, need to link conservation agriculture with residue management, which is elaborated later but need to be emphasized here as well.

Done

---

## Author Comment (AC2) · 6 Mar 2018

Reply to Reviewer 2. Interactive comment on "Challenges of soil carbon sequestration in NENA Region" by Talal Darwish et al. Anonymous Referee #2

This paper addresses a very important topic for soil degradation and food security. This topic could potentially interest a wide audience readership and is well suited to the journal 'SOIL'. However I have several concerns about the paper, which I explain below.

I was rather disappointed by the method used by the authors, which is basically populating large soil map units with mean values taken from profiles. I'm not sure at all it is

relevant and new to do so.

Thank you the Reviewer 2 for your important and deep remarks. The issue of producing a global soil organic carbon map is a priority on the agenda of FAO Global Soil Partnership and the Intergovernmental Technical Panel on Soils (ITPS). Only in December 5 2017, the FAO launched the version 0.1 of the Global Soil OC map of the topsoil (0-30 cm). The subsoil map is still pending securing the soil information from member states. Our article was based on available and accessible digital soil information.

Data on SOC and soil inorganic carbon (SIC) contents in soils of the NENA region were retrieved from the soil database of the FAO-UNESCO digital soil map of the world (DSMW) at 1:5 Million. The database contains 1700 georeferenced soil profiles collected and harmonized from each member state. These were excavated, sampled by horizon, down to the rock, and analyzed in the laboratory according to the standard world accepted methods (FAO, 2007). The soil map was prepared using the topographic map series of the American Geographical Society of New York, as a base, at a nominal scale of 1:5.000.000. Country boundaries were checked and adjusted using the FAO-UNESCO Soil Map of the World, on the basis of FAO and UN conventions. Soil classification was based on horizon designation, depth, texture, slope gradient and soil physico-chemical and chemical properties. Statistical (weighted) average was calculated for the topsoil (0-30 cm) and for the subsoil (30-100 cm) for the full series of chemical and physical parameters sufficient to assess main agricultural soil properties. To fill the gap in some attributes and complete the fields for which no data were available, an expert opinion internationally known soil scientists was used.

I would have expected that at least climate data and land use data are added to the dominant soil type information to do this exercise. I know of course that from a very global point of view, soil types and climate are related but this is not always the case (e.g. you can find rendzinas, fluviosols, cambisols, lithosols, arenosols, etc. in nearly all parts of the World).

The land cover /land use (LCLU) map was produced from ESA information and the soil data were linked to the relevant land cover to show the relevant SOC accumulation as affected by LCLU. The LCLU map is available (beside the SIC maps) and we did not include them because this will multiply the number of maps. But, we will include the LCLU map in the final version of the paper with the surface relevant to each type.

I'm also disappointed by the fact that C1 SOILD Interactive comment Printer-friendly version Discussion paper the authors do not integrate land-use in their mapping, though they show it has a major effect. This seems to me contradictory. I would have expected a more novel approach in mapping such as the use of Digital Soil Mapping. By the way, there have several attempts to map SOC at the global scale and the authors seem to ignore them (e.g. maps from Stockmann et al; Hengl et al; and the recent exercise under the umbrella of the FAO). I would at least compare the results from this study with previous ones in a discussion part.

We used the available and accessible digital soil map of the world done by FAO and UNESCO in 2003 and updated in 2007. This map provide very detailed soil information that can be used in GIS to calculate the SOC and SIC stock (Tons) and density (Ton/ha) in a given country of NENA region. Data on SOC and soil inorganic carbon (SIC) contents in soils of the NENA region were retrieved from the soil database of the FAO-UNESCO digital soil map of the world (DSMW) at 1:5 Million. The database contains 1700 georeferenced soil profiles collected and harmonized from each member state. These were excavated, sampled by horizon, down to the rock, and analyzed in the laboratory according to the standard world accepted methods (FAO, 2007). The soil map was prepared using the topographic map series of the American Geographical Society of New York, as a base, at a nominal scale of 1:5.000.000. Country boundaries were checked and adjusted using the FAO-UNESCO Soil Map of the World, on the basis of FAO and UN conventions. Soil classification was based on horizon designation, depth, texture, slope gradient and soil physico-chemical and chemical properties. Statistical (weighted) average was calculated for the topsoil (0-30 cm) and for the subsoil

(30-100 cm) for the full series of chemical and physical parameters sufficient to assess main agricultural soil properties. To fill the gap in some attributes and complete the fields for which no data were available, an expert opinion internationally known soil scientists was used.

We did not compare the produced map with that of Hengl et al., 2015 (which is also based on point profiles) because the global SOC map in Hengl soilGrids1km layers shows the soil organic carbon content in permille in 0-5 cm (Figure 7A) and the Predicted global distribution of the soil organic carbon stock in tonnes per ha for 0–200 centimetres (Figure 10). In our paper we followed the standard methodology of the measured SOC stock and density in topsoil (0-30 cm) and subsoil (30-100 cm). The first Global SOC Map was launched on December 5, 2017. I contributed to the production of this map based on profile data provided by member states. But this map shows SOC stock in 0-30 cm. Long way to go to complement the whole soil depth until 100 cm. Actually, there is no need to go deeper than 100 cm in the SOC mapping as most crops have their active roots in the layer 20-50 cm (annual crops) and 30-100 cm (fruit trees). But a comparison of values of SOC content (%) and SOC stock revealed comparable values for the C content (1-2%) and stock (density) between 20 and 204 ton/ha. We will add this adapted paragraph to the discussion referring to Hengl T, de Jesus JM, MacMillan RA, Batjes NH, Heuvelink GBM, et al. (2014) SoilGrids1km-Global Soil Information Based on Automated Mapping. PLoS ONE 9(8): e105992. doi:10.1371/journal.pone.0105992. AS for the paper of Stockmann et al we requested a copy from the authors to discuss the two maps.

There is quite no discussion about uncertainty, no error bars or box-plots in results, and this is a serious concern. Validation against Lebanon may induce a serious bias as Lebanon soils have been much more known and investigated for a very long time than soils from some other countries. Subsequently it may be that the DSMW is much more precise in Lebanon than in other regions.

Mapping was done on a small scale, which could be a source of a loss of information.

To test this, the results of the current estimation (1:5 Million) of SOC stocks in Lebanon, were compared with the large scale mapping (1:50,000). This comparison showed discrepancies between 11% for the topsoil and 14% for the subsoil (Darwish and Fadel, 2017). Therefore, the level of uncertainty falls within the admitted diagnostic power of soil mapping, estimated to be close enough to the reported range of map units' purity in reference areas, i.e., a matching between 65% and 70% (Finke et al., 2000). Loss of information related to small, non-mapable soil units in small scale mapping (1:5 Million or 1:1 Million) could be corrected by national and subregional large scale soil mapping (1:50,000 and 1:20,000). The soil data from Lebanon were retrieved from the digital soil map of Lebanon at 1:50,000 georeferenced database produced in 2006 based on 450 excavated, described and analyzed soil profiles. I am the main author of this map. The issue of less soil studies in other NENA countries might be relevant but I visited the portal of the FAO and found 1228 legacy soil maps (http://www.fao.org/soils-portal/en/).

The Mat&Meth section is not enough detailed. We don't know how many profiles were used (a 'large number'), how many per soil type, if they were georeferenced or not and, if so, with which accuracy, when these profiles were taken (they might be no more representative of SOC stocks if they were taken 50 years ago). We also do not know how the calculations by land use were done. By a geographical way, or by extracting soil use classes from the DSMW? There is also here a question of matching observations both in space and time.

Data on SOC and soil inorganic carbon (SIC) contents in soils of the NENA region were retrieved from the soil database of the FAO-UNESCO digital soil map of the world (DSMW) at 1:5 Million. The database contains large number of 1700 georeferenced soil profiles collected and harmonized from each member state. These were excavated, sampled by horizon, down to the rock, and analyzed in the laboratory according to the standard world accepted methods (FAO, 2007). The soil map was prepared using the topographic map series of the American Geographical Society of New York, as a base, at a nominal scale of 1:5.000.000. Country boundaries were checked and

adjusted using the FAO-UNESCO Soil Map of the World, on the basis of FAO and UN conventions. Soil classification was based on horizon designation, depth, texture, slope gradient and soil physico-chemical and chemical properties. Statistical (weighted) average was calculated for the topsoil (0-30 cm) and for the subsoil (30-100 cm) for the full series of chemical and physical parameters sufficient to assess main agricultural soil properties. To fill the gap in some attributes and complete the fields for which no data were available, an expert opinion internationally known soil scientists was used.

Using the DSMW and its updated attribute database maps of the SOC and SIC stock and distribution in 20 NENA states were produced. The scale used in the DSMW is 1:5 Million (FAO, 2007). The soil map was prepared using the topographic map series of the American Geographical Society of New York, as a base, at a nominal scale of 1:5.000.000. Country boundaries were checked and adjusted using the FAO-UNESCO Soil Map of the World, on the basis of FAO and UN conventions. To produce the maps representing the spatial distribution of SOC and SIC, ArcMap 10.3 was used to join the symbology of the C stocks and density with quantities classified into five numerical categories with natural breaks. The explicative note to the DSMW indicates that the data was collected from 1995 up to 2003 when the soil map was first produced. It was thereafter updated with new information in 2007.

The calculation of SOC by land cover /land use was done by intersecting the relative LCLU with the soil type using GIS. The ESA LCLU map was produced in 2016. The soil type is not changed like land use might change.

Finally the general discussion about the challenges is rather interesting but quite fully disconnected from the results.

We modified the discussion in this part to link it more to the challenges and prcatices.

More detailed comments

The legend of Fig 2 is unclear (Mega ton on which area?)

Mega ton per country (the total SOC stock in each country). Fig 2 presents the spatial view of total soil organic carbon stock (mega ton) across the countries of NENA region

The threshold of 30 tons is highly questionable There is twice the same sentence lines 90-92 and lines 101-103.

This threshold is what available in literature. We used it to stress the need to enhance C sequestration in drylands.

There is a problem with the labelling of countries (Fig. 5 top) Corrected

Abstract first word shoul be Near and not North Corrected

---

## Author Comment (AC3) · 8 Mar 2018

First I wish I can correct the Global Soil OC map version 1.0 and not 0.1.

Allow me to add this part added to the manuscript according to your kind suggestion.

Land cover mapping and effect on SOC stock

Land cover map of NENA region shows nearly 80% of the area is covered with bare lands (Figure 4a, b). Grassland, sparse vegetation cover and rainfed agriculture are close by area varying between 4.39% and 5.27%. The irrigated crops do not exceed 0.66% of the total area. Apparently for this reason, The region is becoming increasingly dependent on food imports, because of demographic pressure, rapid urbanization, water scarcity and climate change (FAO, 2015).

[Figure]

Comparing our results with the global SOC map produced by Hengl et al., (2014), based on soilGrids1km layers showing the soil organic carbon content in permille in 0-5 cm and the predicted global distribution of the soil organic carbon stock in tonnes per ha for 0–200 cm to be beyond the followed by FAO methodology of SOC stock estimation and presentation. In this paper the standard methodology of the measured SOC stock and density in topsoil (0-30 cm) and subsoil (30-100 cm) was followed. The first Global SOC Map was launched on December 5, 2017. However, a comparison of values of SOC content (%) and SOC stock revealed comparable trends values for the C content and stock (1-2% and 20-204 ton/ha), with higher upper density in Hengel et al approach.

In our study, the combination of SOC stock map with the land cover map showed significant effect of land cover on SOC stocks in NENA region. As can be expected, Shrublands, sparse vegetation and bare lands gave the smallest values, between 14 and 26 ton ha-1 (Figure 5). In a mixture of shrublands and herbaceous vegetation, the SOC increases to 40 ton ha-1. The highest density (30 and 60 ton ha-1) is found under forest stands.

Despite the expected impact of frequent plowing, the soils under mixed trees and irrigated crops have higher density than rainfed crops. The highest SOC stock was observed under evergreen forest land whose area is very limited (3380 km2 corresponding to 0.02% from the total area). Surprisingly, the stock found under urban soils ($\approx$ 30 tons ha-1) was moderate. This could be related to the urban encroachment on prime soils. Between 1995 and 2015, rapid urban growth caused the loss of over 53 Million tons of soils, 16% of which correspond to prime soils (Darwish and Fadel, 2017). The assessment of SOC content in time and space in relation to land cover showed a decline of OC content in topsoil by up to 1% between 2001 and 2009 (Stockmann et al., 2015). Land cover change was considered as the primary agent that influences SOC change overtime, followed by temperature and precipitation.

Please also note the supplement to this comment:
https://www.soil-discuss.net/soil-2017-39/soil-2017-39-AC3-supplement.pdf

**Supplement:**

[Figure]

*Others: Mixed trees, Neadleaved evergreen trees and Urban

Figure 4a. Proportion of main land cover and land use in NENA region (Source: ESA, 2015)

[Figure]

Figure 4b. Land cover map of NENA region (Source: ESA, 2015
http://maps.elie.ucl.ac.be/CCI/viewer/index.php)

---

## Author Response (AR3)

[revised manuscript text omitted]
 le content of of SOC, as suggested by thein a given soil unit or type that can represent large standard deviations of the means (Figure 41). Soil classification considers several soil forming factors including soil depth, horizon designation and evolution and topographic location. The SOC content can vary depending on soil type location on slopping or level lands, land cover, erosion-sedimentation and soil management. Therefore, the SOC content is the major source of variability forfor the SOC densityy when assessing at the at higher classification levels (soil class level.), with Cambisols and Fluvisols showpresenting the largest standard deviation, caused by a long land use history and large anthropologicanthropogenic impact. Subsoil is less subject to pedo-turbation and direct human influence, thus SOC content has lower variability.

3.3. Land cover mapping and stocks ofeffect on SOC stock

Based on the land cover map of ESA, the bare lands correspond to Land cover map of NENA region shows nearly 80% of the whole NENA regionarea is covered with bare lands (Figure 45a, b). Grassland, sparse vegetation cover and rainfed agriculture representare close by area varying between 4.39% and 5.27%. The irrigated crops do not exceed 0.66% of the total area, distributed in limited usefulcultivated area (Figure 65). The NENA region possesses a land area about 15 million km$^2$, with a total population

exceeding 400 million inhabitants (about 6% of world population) but with only 1% of the world's renewable water resources (https://www.slideshare.net/FAOoftheUN/plenary1-keynote-speech-16dec2013az). Apparently for this reason, TheIne fact that factthe NENA region, represents anthe irrigated area equivalentcorresponds to to 247.5 m$^2$ per capita. This iscould 
[revised manuscript text omitted]

---

## Author Response (AR4)

217-39-Author's_response-version 5

Response to Reviewers showing the latest modifications:

The full names were written as requested.

    1. Line 27-33. A part of the abstract was modified to:

[revised manuscript text omitted]